# Methylglyoxal-Derived Nucleoside Adducts Drive Vascular Dysfunction in a RAGE-Dependent Manner

**DOI:** 10.3390/antiox13010085

**Published:** 2024-01-10

**Authors:** Seigmund Wai Tsuen Lai, Supriyo Bhattacharya, Edwin De Jesus Lopez Gonzalez, Sarah C. Shuck

**Affiliations:** 1Department of Diabetes and Cancer Metabolism, Arthur Riggs Diabetes and Metabolism Research Institute, City of Hope Comprehensive Cancer Center, Duarte, CA 91010, USA; slai@coh.org (S.W.T.L.); elopezgonzalez@coh.org (E.D.J.L.G.); 2Department of Computational and Quantitative Medicine, City of Hope Comprehensive Cancer Center, Duarte, CA 91010, USA; sbhattach@coh.org

**Keywords:** glucose metabolism, glycation, receptor for advanced glycation end products (RAGE), methylglyoxal, diabetes, diabetic kidney disease, endothelial dysfunction, endothelial cells, vascular disease

## Abstract

Diabetic kidney disease (DKD) is a leading cause of death in patients with diabetes. An early precursor to DKD is endothelial cell dysfunction (ECD), which often precedes and exacerbates vascular disease progression. We previously discovered that covalent adducts formed on DNA, RNA, and proteins by the reactive metabolic by-product methylglyoxal (MG) predict DKD risk in patients with type 1 diabetes up to 16 years pre-diagnosis. However, the mechanisms by which MG adducts contribute to vascular disease onset and progression remain unclear. Here, we report that the most predominant MG-induced nucleoside adducts, *N*^2^-(1-carboxyethyl)-deoxyguanosine (CEdG) and *N*^2^-(1-carboxyethyl)-guanosine (CEG), drive endothelial dysfunction. Following CEdG or CEG exposure, primary human umbilical vein endothelial cells (HUVECs) undergo endothelial dysfunction, resulting in enhanced monocyte adhesion, increased reactive oxygen species production, endothelial permeability, impaired endothelial homeostasis, and exhibit a dysfunctional transcriptomic signature. These effects were discovered to be mediated through the receptor for advanced glycation end products (RAGE), as an inhibitor for intracellular RAGE signaling diminished these dysfunctional phenotypes. Therefore, we found that not only are MG adducts biomarkers for DKD, but that they may also have a role as potential drivers of vascular disease onset and progression and a new therapeutic modality.

## 1. Introduction

The rate of diabetes worldwide is expected to double by 2050, raising significant concerns about the risk of developing deadly co-morbidities [1]. Patients with diabetes are at risk of developing microvascular and macrovascular complications, including diabetic kidney disease (DKD), cardiopathy, and neuropathy [2]. These complications are driven by endothelial cell dysfunction (ECD), which can cause impaired endothelial homeostasis, vascular integrity, cell adhesion, tone, and solute transport, all of which can be exacerbated by metabolic changes that occur in patients with diabetes [3]. However, the role of metabolic alterations in driving ECD, and whether they can be used to predict ECD prior to disease onset, remains unknown. Defining these mechanisms would allow for the development of clinical interventions to mitigate or reverse ECD before irreversible damage to the endothelium occurs. Dysregulated endothelial homeostasis can impact disease progression, severity, and treatment approaches, such as in cardiac [4,5], neurological [6], and kidney disorders [7]. In addition to this, metabolic dysfunction is proposed to drive ECD via the production of reactive molecules. An abundant metabolic by-product is methylglyoxal (MG), which covalently modifies macromolecules to form MG-derived adducts (MG adducts) on DNA, RNA, and proteins, namely *N*^2^-carboxyethyl-2′-deoxyguanosine (CEdG), *N*^2^-carboxyethyl-guanosine (CEG), and carboxyethyllysine (CEL), respectively [8]. We recently discovered that nucleoside MG adducts predict DKD at least 16 years pre-diagnosis [9]. MG adducts are a subclass of advanced glycation end products (AGEs), which have been explored in terms of protein modifications, but the role of nucleoside modifications in driving disease is not known [8].

MG adducts are proposed to be indicators of the metabolic flux that often occurs in diseases such as cancer and diabetes. In addition to their role as biomarkers and predictors of DKD, MG adducts may also drive ECD. Here, we provide the first evidence that MG adducts can drive ECD by activating the receptor for AGEs (RAGE), a receptor that has been well studied in various disease pathologies. RAGE is a transmembrane immunoglobulin receptor, with the extracellular V domain proposed to be an important ligand-binding domain that triggers downstream signaling [10,11]. The intracellular RAGE/DIAPH1 interaction has been identified as an important mediator of RAGE signaling [12]. RAGE inhibitor studies targeting this interaction have validated lead compounds as effective inhibitors of RAGE signaling in vitro and in vivo [12]. Examples of ligands identified to bind at the V domain and activate RAGE are the S100B protein, protein-based AGEs, amyloid beta peptide, macrophage antigen-1, lysophosphatidic acid, and the high mobility group box 1 protein [11,13]. Alternatively, S100A6 binds to the C2 domain and activates the JNK and caspase 3/7 apoptotic pathways [14], and S100A12 binds to the C1 and C2 domains, activating the NFκB and MAPK pathways, leading to endothelial activation [15,16]. Compounds such as imidazole, pyrraline, pentosidine, and argypirimidine have also been predicted to act via the C1 domain in in silico docking analyses [17]. However, the literature surrounding these ligand–RAGE interactions is primarily limited to that of large oligomers and macromolecular interactions, namely proteins. For example, studies on other AGEs, such as CEL and carboxymethyllysine (CML), found that while they engage RAGE when bound to larger proteins and peptides, free CEL and CML generally do not interact or activate RAGE [18]. We previously observed a longitudinal association of CEdG and CEG with DKD at least 16 years pre-diagnosis, but their cellular effects are not known [9]. In endothelial cells, RAGE activation has been implicated in endothelial dysfunction, exacerbating inflammation, metabolic dysregulation, and barrier permeability in in vitro and in vivo models of vascular disease [19,20,21]. We recently reviewed the role of MG and MG-derived AGEs in systemic disease [8]. 

Many strategies aimed at treating vascular pathologies fall short in that they do not address the early, underlying metabolic changes that drive disease. As metabolic dysfunction typically precedes vascular disease, this represents a window of opportunity for intervention before disease onset. Thus, there is an urgent need to further our biochemical and metabolic understanding of complication pathogenesis to develop novel therapeutic approaches for diabetic complications. As nucleosides, it is conceivable that MG adducts make chemical interactions with RAGE distinct from other protein AGEs because of their size, charge, thermodynamic interactions, etc. Therefore, it remains unclear whether MG adducts share the same affinities for RAGE in binding, activation, and relation to disease pathology. Furthermore, the phenomenon of nucleoside-mediated RAGE activation and downstream signaling has never been previously studied. Therefore, as by-products of dysregulated metabolism, we thus postulated that MG adducts may induce vascular dysfunction. Here, we provide the first biochemical evidence that nucleoside MG adducts can drive ECD mediated by RAGE activation and signaling. 

## 2. Materials and Methods

### 2.1. Cell Cultures

Primary human umbilical vein endothelial cells (HUVECs) and THP-1 monocytic cells were a generous gift from Dr. Zhen Chen at City of Hope. HUVECs were cultured in M199 media (Sigma Aldrich, St. Louis, MO, USA) supplemented with 10% heat-inactivated FBS (Omega Scientific, Tarzana, CA, USA) and β-endothelial cell growth factor (Sigma Aldrich, St. Louis, MO, USA). HUVECs between passages 5 and 9 were used for the experiments. THP-1 cells were cultured in RPMI-1640 (ThermoFisher Scientific, Waltham, MA, USA) supplemented with 10% heat-inactivated FBS (Omega Scientific, Tarzana, CA, USA). The cells were maintained in an incubator at 37 °C with 5% CO_2_ and were routinely checked for mycoplasma contamination. Bovine serum albumin was conjugated to sodium palmitate as previously described, and an unconjugated vehicle was prepared in parallel [22].

### 2.2. Materials

RAGE inhibitor (Ri), (4-[(4-bromobenzyl)amino]phenol), was purchased from ChemBridge (ID# 7737211; San Diego, CA, USA) [12]. CEdG and CEG were synthesized as previously described using deoxyguanosine monophosphate (dGMP) and guanosine monophosphate (GMP), respectively [23,24]. ^15^N_5_-*R*,*S*-CEdG and ^15^N_5_-*R*,*S*-CEG were synthesized similarly using ^15^N_5_-dGMP and ^15^N_5_-GMP, respectively. The following chemicals were obtained for our study: Tween-20 and trichloroacetic acid from ThermoFisher Scientific, Waltham, CA, USA; phosphate-buffered saline (PBS) from Bioland Scientific, Paramount, CA, USA; and Triton X-100, trypan blue, crystal violet, and endotoxin-free fatty acid-poor bovine serum albumin (BSA) from Sigma Aldrich, St. Louis, MO, USA, unless otherwise indicated.

### 2.3. Multiplex Mass Spectrometry Method to Measure MG Adducts

MG adducts were isolated and quantified via liquid chromatography–tandem mass spectrometry (LC-MS/MS) as previously described, with data analysis performed using MassHunter quantitative analysis software (Agilent Technologies, Santa Clara, CA, USA; Version B.07.01, Build 7.1.524.0) [9,23,24].

### 2.4. Isolation and Enrichment of Secreted MG Adducts

The cells were treated as indicated, and the cultured media was harvested, with cellular debris pelleted through centrifugation for 5 min at 1250× *g* and processed through 3-kDa filters (ThermoFisher Scientific, Waltham, MA, USA). The flowthrough was subjected to cation exchange solid-phase extraction using Waters™ Oasis MCX cartridges as described previously [9,23,24]. Samples were dried at room temperature using vacuum centrifugation and resuspended in 25 µL of LC-MS-grade water (Fisher Scientific, Waltham, MA, USA). 

### 2.5. Experimental Treatments

Unless otherwise indicated, HUVECs were exposed to MG adducts at a final concentration of 50 ng/mL of each diastereomer (*R* and *S*). Where indicated, HUVECs were pretreated with Ri at a final concentration of 100 µM for 1 h before treatment with MG adducts. 

### 2.6. RNA Extraction and Quantitative PCR

Total RNA was isolated using Direct-zol RNA Miniprep Plus (Zymo Research, Irvine, CA, USA). cDNA was synthesized using the iScript™ Reverse Transcription Supermix (Bio-Rad, Hercules, CA, USA). qPCR was performed with the PowerUp™ SYBR™ Green Master Mix (Applied Biosystems, Foster City, CA, USA) using the ViiA™ 7 Real-Time PCR system (Applied Biosystems, Foster City, CA, USA). All primer sequences used in qPCR amplification can be found in Appendix A.

### 2.7. Crystal Violet Viability Assay

HUVECs were plated in 96-well plates at a density of 21,000 cells/cm^2^ and treated the next day with the indicated concentrations of each compound or vehicle for 24 h. The media was decanted, and the wells were washed with 100 µL of PBS to remove dead cells. The cells were stained with 100 µL of crystal violet solution (0.5% crystal violet (*w*/*v*) and 25% methanol (*v*/*v*) in water) for 30 min with gentle rocking. The crystal violet solution was removed, and the wells were washed with 100 µL of PBS. The plates were airdried for 30 min at room temperature, and 100 µL of 1% SDS was added to solubilize the stain. Absorbance at 562 nm was read using a Cytation 5 cell imaging multi-mode reader (BioTek Instruments, Winooski, VT, USA). The experiments were performed with three independent biological replicates per condition. Treatment survival percentage was normalized to the vehicle untreated control. IC_50_ values were calculated using GraphPad Prism software (Version 9.5.1; Boston, MA, USA).

### 2.8. Proliferation and Morphology Assay

HUVECs were plated in 24-well plates at a density of 16,000 cells/cm^2^ and treated the next day with the indicated concentrations of each compound or vehicle for the designated timepoints. At each timepoint (day 1, 3, or 5), cells were imaged using EVOS™ XL Core (ThermoFisher Scientific, Waltham, MA, USA), harvested, and live cells were counted using the LUNA-II™ cell counter (Logos Biosystems, Anyang-si, Republic of Korea) using trypan blue exclusion.

### 2.9. Immunofluorescence

HUVECs were plated in 24-well plates on coverslips at a density of 26,000 cells/cm^2^ (NFκB staining) or 38,000 cells/cm^2^ (MG adduct staining) and allowed to adhere overnight and then treated as indicated. The cells were washed with ice-cold PBS and fixed with ice-cold 100% methanol for 20 min at −20 °C. The fixed cells were rinsed once with ice-cold PBS for 5 min with gentle shaking. The cells were then permeabilized for 10 min on ice with 0.3% Triton X-100 in PBS. The cells were rinsed three times with ice-cold PBS for 5 min each with gentle shaking and incubated with blocking buffer (1% BSA in PBS) for 1.5 h at room temperature. Mouse anti-human NFκB p-65 antibody (Cell Signaling Technology, Danvers, MA, USA; 6956, 1:400) or mouse anti-MG adduct antibody (Cell BioLabs, San Diego, CA, USA; STA-011, 1:200) was added to the coverslips in PBS containing 1% BSA and 0.1% Tween-20 and incubated overnight at 4 °C with gentle rocking. The next day, the coverslips were washed with 0.1% Tween-20 in PBS three times for 5 min with gentle shaking. The coverslips were incubated with Alexa Fluor 488 (AF488)-conjugated goat anti-mouse antibody (Fisher Scientific, Waltham, MA, USA; A-11001, 1:1000) in PBS containing 1% BSA and 0.1% Tween-20 for 1 h at room temperature. The coverslips were washed with 0.1% Tween-20 in PBS three times for 5 min with gentle shaking and mounted onto slides with Fluoroshield mounting medium (Fisher Scientific, Waltham, MA, USA). Slides were sealed with nail polish and allowed to cure for at least 2 h. Fluorescent images were taken with a Zeiss Observer II brightfield microscope with a 20X objective and analyzed using Zeiss ZEN Lite software (Version 3.7.97.01000; Oberkochen, Germany). Nuclear NFκB translocation was determined by outlining the DAPI-positive nuclei and quantifying the AF488 signal in the nucleus. MG adduct quantification was performed by outlining each cell and quantifying the intracellular AF488 signal.

### 2.10. Immunoblotting 

HUVECs were treated as indicated and lysed using RIPA lysis buffer (20 mM Tris-HCl (pH 7.5), 150 mM NaCl, 1 mM Na_2_EDTA, 1 mM EGTA, 1% NP-40, 1% sodium deoxycholate, 2.5 mM sodium pyrophosphate, 1 mM beta-glycerophosphate, 1 mM Na_3_VO_4_, 1 µg/mL leupeptin, 1 mM PMSF, and 1.25 mM DL-dithiothreitol) supplemented with a Pierce protease/phosphatase inhibitor cocktail (Thermo Fisher Scientific, Waltham, MA, USA). The samples were centrifuged at 14,000× *g* for 15 min at 4 °C. The supernatant was collected and aliquoted, and protein content was determined using the Pierce™ BCA protein assay kit (Thermo Fisher Scientific, Waltham, MA, USA). Proteins (15 µg) were resolved on 4–20% SDS–PAGE gels (Bio-Rad, Hercules, CA, USA) and transferred to PVDF membranes (Bio-Rad, Hercules, CA, USA). The membranes were blocked for 30 min using EveryBlot blocking buffer (Bio-Rad, Hercules, CA, USA) and probed for phospho-AKT (Cell Signaling Technology, Danvers, MA, USA; 4060, 1:1000), AKT (Cell Signaling Technology, Danvers, MA, USA; 4691, 1:1000), phospho-MEK (Cell Signaling Technology, Danvers, MA, USA; 2338, 1:1000), MEK (Cell Signaling Technology, Danvers, MA, USA; 8727, 1:1000), phospho-eNOS (Cell Signaling Technology, Danvers, MA, USA; 9571, 1:1000), eNOS (Cell Signaling Technology, Danvers, MA, USA; 32027, 1:1000), RAGE (Cell Signaling Technology, Danvers, MA, USA; 6996, 1:1000), or GAPDH (Cell Signaling Technology, Danvers, MA, USA; 5174, 1:5000) diluted in EveryBlot blocking buffer (Bio-Rad, Hercules, CA, USA). The blots were then washed three times with PBS + 0.1% Tween-20 for 5 min with gentle rocking. For the secondary antibody, goat anti-rabbit IgG H&L-HRP conjugate (Abcam, Waltham, MA, USA; ab6721, 1:5000) was diluted in EveryBlot blocking buffer (Bio-Rad, Hercules, CA, USA) and incubated for 1 h at room temperature covered from light. The blots were then washed three times with PBS + 0.1% Tween-20 for 5 min with gentle rocking. The blots were incubated with an enhanced chemiluminescent substrate (ThermoFisher Scientific, Waltham, MA, USA) and imaged using the iBright™ FL1500 imaging system (Invitrogen, Carlsbad, CA, USA).

### 2.11. Monocyte Adhesion Assay

HUVECs were plated in 24-well plates at a density of 20,000 cells/cm^2^ and allowed to adhere overnight. HUVECs were then treated as indicated. THP-1 monocytic cells were labeled with 200 nM of calcein AM (Invitrogen, Carlsbad, CA, USA; C1430). THP-1 cells were incubated with HUVECs for 30 min at 37 °C. Unbound monocytes were removed by washing with a complete HUVEC medium. Fluorescent and attached monocytes were imaged using a Cytation 5 cell imaging multi-mode reader (BioTek Instruments, Winooski, VT, USA) with the green fluorescent channel. 

### 2.12. Reactive Oxygen Species (ROS) Detection

HUVECs were plated in 96-well plates at a density of 21,000 cells/cm^2^, allowed to adhere overnight, and then treated as indicated. ROS detection was performed according to the manufacturer’s instructions (Cayman Chemical, Ann Arbor, MI, USA; 601290). The plate was read using a Cytation 5 cell imaging multi-mode reader (BioTek Instruments, Winooski, VT, USA). Antimycin A (“Anti”) was used as a positive control to induce ROS formation.

### 2.13. Transwell Endothelial Permeability Assay

Endothelial permeability was determined using the endothelial transwell permeability assay (Cell Biologics, Chicago, IL, USA; CB6929). HUVECs (200,000 cells per well) were seeded onto 24-well cell culture inserts provided with the kit and grown to a confluent monolayer. The media was then removed from each well and replaced with serum-free media. The cells were then treated as indicated, with or without the 1-h pre-treatment with Ri (100 µM). HRP (5 µL, 44 kDa) was then added to the top chamber of inserts, and the cells were incubated at 37 °C with 5% CO_2_. At each designated timepoint, 20 µL aliquots of the medium in the bottom chamber were transferred to a 96-well plate in triplicate. Permeability was assessed, according to the manufacturer’s instructions, by measuring sample absorbance at 450 nm using a Cytation 5 cell imaging multi-mode reader (BioTek Instruments, Winooski, VT, USA). The data were presented as relative permeability relative to untreated control values. The experiments were performed in biological triplicates.

### 2.14. Measuring MG Adduct Cellular Uptake

HUVECs were plated on 6-well plates at a density of 30,000 cells/cm^2^ and treated the next day with 200 ng/mL of isotopically labeled MG adducts (^15^N_5_-*R*,*S*-CEdG or ^15^N_5_-*R*,*S*-CEG). At each indicated timepoint, conditioned media was collected, centrifuged to remove cellular debris, and processed via cation exchange solid-phase extraction using Oasis MCX SPE cartridges (Waters, Milford, MA, USA) [9,23,24]. Additionally, cell pellets were collected and lysed using 10% *w*/*v* trichloroacetic acid, and cellular debris was separated via centrifugation for 15 min at 4 °C at 12,000*× g*. Both the cell pellets and conditioned media were spiked with isotopically labeled internal standards prior to their quantification via LC-MS as previously described [23,24]. To quantify adduct uptake, cells treated with ^15^N_5-_*R*,*S*-CEdG used ^15^N_5_-*S*-CEG at a final concentration of 5 ng/mL as an internal standard. Conversely, cells treated with ^15^N_5-_*R*,*S*-CEG used ^15^N_5_-*S*-CEdG at a final concentration of 5 ng/mL as an internal standard. 

### 2.15. RNA Sequencing

Total RNA was isolated using Direct-zol RNA Miniprep Plus (Zymo Research, Irvine, CA, USA). RNA sequencing libraries were generated with the Kapa RNA HyperPrep kit with RiboErase (Kapa Biosystems, Wilmington, MA, USA; KR1351), according to the manufacturer’s protocol. Total RNA (250 ng) from each sample was used for sequencing library preparation. The final libraries were validated with the Agilent Bioanalyzer High Sensitivity DNA kit (Agilent Technologies, Santa Clara, CA, USA). Sequencing was performed on the Illumina NovaSeq6000 with S4 Reagent Kit v1.5 (Illumina, San Diego, CA, USA; 20028313) with a sequencing length of 2 × 101. Real-time analysis (RTA) 3.4.4 software was used to process the image analysis.

### 2.16. Bioinformatics Analysis

The RNA-seq fastq reads were processed using Partek Flow Genomics pipeline version 10.0 [25]. The key steps involved in this pipeline are given below. Raw fastq reads were preprocessed to filter low-quality reads and contaminants using Bowtie 2 [26] and remove adapters using cutadapt [27]. The cleaned reads were aligned to the reference genome (hg38) using STAR version 2.7.8a [28]. Gene counts were quantified using a modified form of the expectation/maximization algorithm (Partek E/M) [29]. Count normalization and differential gene expression (DEG) analyses were performed using limma-voom [30]. The normalized gene counts were converted into counts per million (CPM), followed by the filtering of low-expressed genes (genes with CPM < 0.3 in more than 50% of the samples). *p*-values of the DEGs were converted into false discovery rates (FDRs) according to the Benjamini–Hochberg method [31]. Gene set enrichment analysis (GSEA) was performed using ClusterProfiler [32] using Gene Ontology biological process (GO-BP) [33,34], Kyoto Encyclopedia of Genes and Gene Products (KEGG), and Molecular Signatures Database (MsigDB) Hallmark terms [35,36]. For calculating gene set enrichment, genes were ranked according to their −log(pvalue). Protein interaction analysis was performed using STRING by employing the STRINGdb R package [37]. The volcano plots, heatmaps, and Venn diagrams were created using the EnhancedVolcano [38], ComplexHeatmap [39], and eulerr packages, respectively [40]. Protein network diagrams were created using the igraph R package [41]. The analysis pipeline was implemented in R version 4.2 [42]. The data presented in this study are available in the National Center for Biotechnology Information (NCBI)’s Gene Expression Omnibus (GEO), under accession number GSE251645.

### 2.17. Statistical Analysis

Statistical analyses were conducted using GraphPad Prism Software (Version 9.5.1; Boston, MA, USA). Gaussian distribution was determined to evaluate normality. Comparisons between two groups were carried out with the unpaired *t*-test. One-way ANOVA with Dunnett’s or Tukey’s multiple comparisons test was used to compare two or more groups. Data were presented as the mean ± SD; *p* < 0.05 values were considered statistically significant.

## 3. Results

### 3.1. Endothelial Cells Produce and Secrete MG Adducts under Hyperglycemic and Hyperlipidemic Diabetic Conditions

We sought to determine the impact of diabetogenic conditions on the intracellular accumulation and extracellular release of MG adducts in HUVECs. These cells were treated with D-glucose (5.5 mM in media, “Untreated”; 25 mM, “Glu”), palmitic acid (100 μM, “PA”) conjugated to bovine serum albumin (BSA), or a combination of the two (Glu + PA) for 24 h. Basal glucose levels in the media were 5.5 mM, and unconjugated BSA vehicle treatment was included as a control (“BSA”). A schematic representing the experimental design can be found in Figure 1A. Following treatment, MG adduct levels were quantified using immunofluorescence with an antibody that recognizes DNA, RNA, and protein MG adducts. Treatment with Glu or PA both significantly increased intracellular MG adduct levels, while the combination Glu + PA treatment resulted in higher intracellular levels of MG adducts than either alone (Figure 1B,C). We next measured the excretion of MG adducts from HUVECs treated with these diabetogenic conditions. Conditioned media was harvested, and MG adducts were quantified using stable isotope dilution LC-MS/MS (as outlined in the Materials and Methods section) (Figure 1A) [23,24,43]. Extracellular CEdG and CEG levels were significantly elevated in HUVECs treated with either Glu or PA but not in cells treated with the BSA vehicle control (Figure 1D,E). Furthermore, combination treatment with Glu + PA increased secreted CEdG and CEG beyond what was observed with either treatment alone (Figure 1D,E).

### 3.2. MG Adducts Are Not Toxic to HUVECs 

To first determine the impact of MG adducts on HUVEC viability, proliferation, and morphology, *R*,*S*-CEdG and *R,S-*CEG adducts were synthesized and purified (Appendix A). We first extrapolated clinically observed urinary concentrations of MG adducts in patients with type 1 diabetes to rationalize our treatment approach [9]. HUVECs were treated with increasing concentrations of these analytes for 24 h, and their viability was determined using crystal violet analysis. We found that CEdG and CEG treatments up to 200 ng/mL were not cytotoxic (Appendix A). To determine whether there were long-term effects on HUVEC proliferation and morphology, we then treated HUVECs with CEdG or CEG at doses up to 200 ng/mL for up to 5 days and monitored their proliferation and morphology. No discernible effects on proliferation or morphology were observed (Appendix A).

### 3.3. MG Adducts Promote Monocyte Adhesion

As by-products of dysregulated metabolism, we proposed that MG adducts may be a driver of vascular disease. An early precursor of vascular disease is endothelial dysfunction, which typically begins with endothelial activation, characterized by increased leukocyte diapedesis, in which immune cells, such as monocytes, migrate and extravasate into the endothelium where they recruit additional immune cells and can induce inflammation [44]. We began by studying two prominent surface adhesion markers that facilitate monocyte adhesion, *ICAM1* and *VCAM1*. Treatment with CEdG led to an approximate 4-fold and 3-fold increase in the expression levels of *ICAM1* and *VCAM1*, respectively (Figure 2A), while CEG led to an approximate 3-fold increase in the expression levels of both (Figure 2B). We next examined the impact of MG adducts on monocyte adhesion and found that the treatment of HUVECs with either CEdG or CEG significantly increased the adherence of THP-1 monocytes to a HUVEC monolayer (Figure 2C,D). The increase in *ICAM1* and *VCAM1* expression and monocyte adhesion were not observed when HUVECs were treated with the MG adducts’ unmodified nucleoside counterparts—deoxyguanosine (dG) or guanosine (G) (Appendix A).

### 3.4. MG Adducts Induce Oxidative Stress and Inflammation

We next examined the impact of the MG adducts on ROS formation, a driver of inflammation. CEdG and CEG, but not dG or G, both led to a significant upregulation of ROS abundance, namely superoxides and peroxides (Figure 3A and Appendix A). To further test whether inflammation was increased following the MG adduct treatment, qPCR analysis of the inflammatory genes *TNFa, IL1b, IL6,* and *IFNg* was performed. We found that both CEdG and CEG significantly increased the expression of these genes (Figure 3B,C), an effect absent with dG or G (Appendix A).

### 3.5. MG Adducts Induce Endothelial Dysfunction and Impair Endothelial Homeostasis and Vascular Integrity

Maintenance of endothelial homeostasis is key to protecting the endothelium’s integrity and health. To determine the impact of MG adducts on endothelial health, we measured eNOS expression and phosphorylation, a key regulator of endothelial homeostasis. MG adducts increased *ENOS* mRNA expression (Figure 4A,B), which may be a cellular response to the stress induced by MG adducts. However, MG adducts significantly decreased eNOS protein expression and phosphorylation, suggesting that they impair the functional component of this pathway (Figure 4C). We did not observe a change in *ENOS* mRNA expression or total and phosphorylated eNOS protein expression in cells treated with dG or G (Appendix A), indicating that these defects are unique to MG adducts.

To determine the impact of MG adducts on endothelial permeability, we then used an HRP probe, which allows for the evaluation of changes in endothelial permeability in real time, determined by the amount of HRP that flows through the endothelial monolayer. MG adducts significantly increased HUVEC permeability at 6 h of treatment, suggesting that MG adducts compromise vascular integrity and permeability, which over time, if left unregulated, could have systemic detrimental effects (Figure 4D).

It has been recently established that *LINC00607* (“607”) is an important regulator and mediator of endothelial dysfunction [45]. The suppression of 607 ablated dysfunctional phenotypes brought on by stressors such as high glucose and TNFα treatment in HUVECs, supporting the role of 607 in mediating endothelial activation and dysfunction [45]. In our studies, we found a 3-fold increase in 607 expression in HUVECs treated with MG adducts (Figure 4A,B). A similar increase in expression was noted in thrombospondin-1 (TSP1 or THBS1), a protein whose receptor promiscuity results in multiple binding partners, the culmination of which contribute to numerous cellular effects, including impaired angiogenesis, increased endothelial apoptosis, increased senescence, and further ROS production (Figure 4A,B). Neither of these changes were observed with dG or G treatment (Appendix A).

### 3.6. Cells Do Not Uptake MG Adducts

We next sought to determine whether these effects were due to endothelial cells internalizing and uptaking MG adducts from the extracellular space. HUVECs were treated with isotopically labeled adducts, namely ^15^N_5-_*R*,*S*-CEdG or ^15^N_5_*-R*,*S*-CEG, for 24 h, and their uptake was measured by harvesting the culture media and cell pellet and quantifying the analyte levels in each compartment using stable isotope dilution LC-MS/MS (Figure 5A). We found that the isotopically labeled MG adducts remained in the conditioned media following 24 h of treatment, with <0.05% found in the cell pellet (Figure 5B). This supports a potential mechanism of action by which the impact of MG adducts on intracellular signaling processes and EC homeostasis is a result of an extracellular interaction. 

### 3.7. MG Adducts Activate RAGE

As a subclass of AGEs, we hypothesized that nucleoside MG adducts may be exerting their effects on endothelial cells via RAGE, an interaction that has not been previously reported. We found that MG adducts significantly increased the phosphorylation of targets downstream of RAGE, namely AKT (3–3.5 fold) and MEK (1.2–1.5 fold) (Figure 6A,B) [46,47,48,49,50,51]. CEdG, CEG, dG, or G treatment of HUVECs did not impact RAGE mRNA or protein levels (Appendix A), and treatment with dG or G did not induce the phosphorylation of AKT or MEK (Appendix A). To determine whether this increase in phosphorylation is RAGE dependent, a small molecule inhibitor of RAGE (“Ri”) known as 4-[(4-bromobenzyl)amino]phenol, identified by Manigrasso et al., was used [12]. Ri targets the intracellular, cytoplasmic tail of RAGE and blocks its interaction with DIAPH1. By targeting this domain, the impacts on other RAGE variants (sRAGE and esRAGE) are avoided. 

We began by characterizing the effects of Ri on HUVEC viability, proliferation, and morphology, as well as RAGE expression. The IC_50_ value of Ri in HUVECs was found to be 320 µM following 24 h of treatment (Appendix A), and Ri did not impact RAGE gene or protein expression (Appendix A). HUVECs were then treated with increasing doses of Ri, and their cell growth and morphology were monitored for 5 days. Up to day 3, there was no impact on HUVEC proliferation or morphology (Appendix A). However, by day 5 of Ri, approximately 50% of cells had become detached from the well, and their morphology had changed to a rounded, flattened shape, with aggregates and cellular debris noted in the media from detached and/or dead cells (Appendix A). 

The pre-treatment of HUVECs with Ri for 1 h prior to the MG adduct treatment suppressed RAGE activation and reduced AKT and MEK phosphorylation to basal levels (Figure 6A). However, the Ri treatment alone did not induce AKT or MEK phosphorylation (Appendix A). This suggests that CEdG and CEG are a novel class of RAGE ligands that are capable of activating RAGE and its downstream cascades. 

### 3.8. MG Adducts Drive NFκB p65 Nuclear Translocation in a RAGE-Dependent Manner

To characterize the impact of CEdG and CEG on pathways downstream of AKT and MEK, NFκB p65 nuclear translocation, an indicator of NFκB activation, was measured. Both RAGE and NFκB activation have been previously implicated in numerous disease pathologies, including vascular complications. After 1 h of CEdG or CEG treatment, nuclear NFκB p65 levels were significantly increased (Figure 6C,D). This was ablated by pre-treatment with Ri (Figure 6C,D), which alone did not elicit NFκB p65 translocation (Appendix A). Furthermore, the unmodified nucleosides dG and G did not lead to NFκB p65 nuclear translocation (Appendix A), suggesting that MG adducts play a key role in mediating NFκB activation and its downstream cellular effects in a RAGE-dependent manner. 

### 3.9. RAGE Inhibition Mitigates MG Adduct-Mediated Endothelial Dysfunction

We then sought to determine whether MG adduct-mediated endothelial dysfunction was the result of RAGE activation and NFκB signaling. To accomplish this, HUVECs were pre-treated with Ri first, prior to the treatment with the MG adducts. It is important to note that the Ri treatment alone did not induce dysfunction (Appendix A). However, we found that Ri pre-treatment before MG adducts strongly reduced endothelial activation and monocyte adhesion (Figure 7A–D), though not entirely. A similar observation was made in ROS production and inflammation (Figure 7E–G). Notably, the Ri pre-treatment of HUVECs also suppressed the expression of *ENOS*, *LINC00607,* and *TSP1* (Figure 7H,I) and prevented MG adduct-induced endothelial dysfunction, evaluated by endothelial homeostasis via eNOS (Figure 7J) and vascular permeability (Figure 7K). Taken together, this demonstrates that MG adduct-induced endothelial dysfunction is mediated through RAGE signaling.

### 3.10. MG Adducts Induce Transcriptional Changes in HUVECs

To understand the transcriptomic-wide effects of MG adducts on HUVECs in relation to endothelial function and biology, we performed RNA sequencing on cells treated with vehicle or either MG adducts with or without Ri. Principal component analysis (PCA) revealed that triplicate samples showed similar clustering and a clear discrimination between groups treated with either MG adducts without Ri (“CEdG” or “CEG”), untreated samples (“Control”), those that were treated with both MG adducts and Ri (“CEdG+Ri” or “CEG+Ri”), or Ri alone (“Ri”). The eigenvalues of the first two principal components accounted for 36% of the total variance (Dim1: 7%; Dim2: 29%) (Figure 8A). The analysis of the differentially expressed genes (DEGs) differentially impacted by CEdG or CEG or shared differences revealed that with a threshold of *p* < 0.1 and log_2_FC of ±0.5, 345 genes were unique to CEG, 2103 genes were unique to CEdG, and 359 genes were differentially expressed between the groups (Figure 8B). 

We next analyzed differences in gene expression and found that several genes involved in EC function and inflammation were significantly up- and down-regulated in cells treated with either CEdG or CEG; a selection of these genes can be found annotated in Figure 8C,D. Interestingly, within this selection, both CEdG and CEG impacted the same genes to different degrees, suggesting that they may act through different mechanisms or preferentially activate different cascades. We then performed gene ontology biological process (GO-BP) and hallmark pathway analyses to define the extent to which these genes and pathways were differentially regulated. From this, we observed a RAGE-dependent upregulation of pathways associated with cell death, inflammation, cell stress, metabolism, extracellular matrix (ECM) remodeling, and angiogenesis (Figure 8E). In the same analysis, we also found a RAGE-dependent downregulation of pathways associated with the cell cycle, developmental biology, endothelial function, genomic stability, protein processing, and cell junctions (Figure 8E). The pre-treatment with Ri ablated these changes, and the Ri treatment alone also resulted in transcriptomic signatures resembling that of the control, untreated groups (Figure 8E). 

We then further dissected the DEGs that were significantly up- or down-regulated by both CEdG or CEG. Taking the genes from our GO-BP pathway analysis, we began by evaluating the genes whose expression was significantly impacted through RAGE by either CEdG or CEG separately (Figure 9A). Both CEdG and CEG induced an increase in the expression of the genes involved in an inflammatory response (e.g., *SERPINE1* and *C2CD4B*), induction of cell death (e.g., *BAD* and *TRADD*), defects in endothelial function (e.g., *NOSIP*, *SIGIRR*, and *EDN1*), and ECM remodeling (e.g., *MMP28*, *SCX*, and *ADAM15*). Conversely, both CEdG and CEG induced a decrease in the expression of the genes involved in cell cycle progression (e.g., *TTK* and *KIF14*), cell junction maintenance (e.g., *CDH6* and *GJC1*), and protection of ECs from dysfunction and stress (e.g., *SIRT1*, *APPL1*, and *MIR17HG*). A heatmap showcasing a selection of genes significantly impacted by both CEdG and CEG is depicted in Figure 9A, an effect found to be mediated through RAGE signaling. 

To understand the transcriptomic effects unique to either adduct, we conducted analyses aimed at studying significant changes from the GO-BP analysis observed only in cells treated with either CEdG or CEG alone. CEdG significantly upregulated the expression of the genes involved in inflammation (e.g., *CCL2* and *TRAF1*), ECM remodeling (e.g., *COL4, COL8,* and *FBLN2*), TGF-ß signaling (e.g., *TGFB1*), and regulators of EC function and integrity (e.g., *PLCB2, UCP2,* and *EMILIN1)* (Figure 9B). CEdG was found to significantly downregulate genes involved in maintaining genomic stability (e.g., *RMI1*), protection against oxidative stress (e.g., *SOD2*), ECM adhesion (e.g., *CD44*), and regulation of EC homeostasis (e.g., *CAV1* and *CAV2*) (Figure 9B). The GO-BP pathway analysis for CEdG-treated samples can be found in Appendix A and revealed positive enrichment for the pathways associated with angiogenesis, metabolism, inflammation, and cell death and negative enrichment for the pathways associated with cell cycle progression and protein processing. The hallmark pathway analysis revealed positive enrichment for the pathways associated with p53 signaling, reactive oxygen species responses, MYC signaling, apoptosis, and oxidative phosphorylation and negative enrichment for the pathways associated with protein secretion and mitosis, an observation recapitulated in our GO-BP analysis (Figure 9C).

CEG was found to significantly upregulate the genes associated with promoting endothelial invasiveness (e.g., *CLIC3*) and regulators of inflammation and cell death (e.g., *RPS7*, *PPIA*, and *PTPMT1*), and downregulate the genes involved in the cell cycle (e.g., *USP22*, *KIF3B*, and *DYNC1H1*), maintenance of the ECM (e.g., *FN1*), endothelial function (e.g., *SMAD3, SMAD6,* and *TET1)*, and potential regulators of TGF-ß signaling (e.g., *FBN1* and *FBN2*) (Figure 9D). The GO-BP pathway analysis for samples treated with CEG can be found in Appendix A, and revealed positive enrichment for the pathways associated with metabolism, stress response signaling, apoptotic signaling, and inflammation, and negative enrichment for the pathways associated with cell cycle progression, organ development, and adherens junctions. The hallmark pathway analysis revealed positive enrichment for the pathways associated with reactive oxygen species responses, MYC signaling, p53 signaling, apoptosis, and immune responses as well. This analysis also identified negative enrichment for the pathways associated with Wnt/ß-catenin signaling, TGF-ß signaling, and cell cycle progression (Figure 9E).

Finally, we conducted protein–protein interaction network analyses to study how these DEGs interact at the protein level. Based on known protein–protein interactions (PPIs) from the STRING database, we constructed an interaction network consisting of all the DEGs (1001 genes in total) that were involved in key biological processes relevant to RAGE-dependent signaling in either CEdG- or CEG-treated HUVECs (Figure 10A). We found that these genes form a strongly interconnected PPI network (*p* = 0), suggesting that changes in these PPIs will cause a coordinated change in the cellular phenotype. Furthermore, large fractions of upregulated and downregulated genes clustered separately in this network (indicated by the blue and red genes in Figure 10A), forming their own separate subnetworks. This suggests a coordinated regulation of the upregulated and downregulated biological processes through separate PPI networks. 

To investigate the differentially regulated PPI network more closely, we selected a smaller set of key DEGs involved in various endothelial functions and RAGE pathways and visualized their mutual protein-level interactions (Figure 10B). The genes that were common and unique to CEG and CEdG were highlighted in different colors. CEdG significantly impacted genes such as those involved in inflammation (e.g., *CCL2*, *CXCL11*, and *UCP2*), maintenance of endothelial function (e.g., *CAV1*, *CAV2*, and *NOS3*), and ECM remodeling (e.g., *MMP28*); CEG significantly impacted genes such as those implicated in maintaining vascular integrity and homeostasis (e.g., *SMAD3* and *SMAD6*), and inflammation (e.g., *IL6*). To further understand the significance of these gene interactions, we clustered this network using the STRING database’s built-in algorithm, resulting in three subnetworks. Each of these subnetworks were found to regulate distinct biological processes, such as the cell cycle, ECM remodeling, and inflammation, as shown in Figure 10B. In summary, the architecture of the PPI network rationalizes the diverse functional impact of RAGE-dependent signaling initiated by the two MG adducts.

### 3.11. Time-Dependent Effects of MG Adducts on Endothelial Cells

We then performed a time-course treatment of MG adducts to gain an understanding of the time-dependent effects of MG adducts on endothelial cells. CEdG and CEG were found to have varying effects on the expression of the endothelial activation markers *ICAM1* and *VCAM1*, which peaked at 1 h of treatment, with a gradual decline in expression occurring thereafter (Appendix A). We also observed a significant reduction in *ENOS* expression following 3 h of treatment with either CEdG or CEG (Appendix A). A similar finding was made in the expression of the dysfunction regulator *LINC00607* (Appendix A). Interestingly, the peak in the expression level of *TSP1* was observed after 3 h of treatment for both CEdG and CEG (Appendix A). The evaluation of inflammatory cytokine gene expression revealed an aberrant expression pattern that did not reveal a clear trend of increase or decrease; however, the minimum 1-h treatment was sufficient to induce a significant increase in their expression, which was followed either by a further increase later or a gradual decrease (Appendix A). We also found that eNOS total expression and phosphorylation remain diminished for up to 3 h of MG adduct treatment, but this is reversed by 6 h, resulting in the significant phosphorylation and expression of eNOS occurring after 6 h of MG adduct treatment (Appendix A).

To study whether RAGE activation by MG adducts was time dependent, we studied the phosphorylation of AKT and MEK over time. Our earlier finding indicated that 1 h of adduct treatment resulted in a 3–4 fold and 1.5–2 fold increase in the phosphorylation of AKT and MEK, respectively (Figure 2A). Our time course study suggests that while AKT mounted an acute phosphorylation response to MG adducts within 1 h, this effect was transient and was no longer present by 3 h of treatment (Appendix A). This is contrasted by MEK, which did not demonstrate as strong a phosphorylation response to the 1-h MG adduct treatment. The phosphorylation of MEK was sustained beyond 1 h and increased in a time-dependent manner, reaching a peak around 3 h, and a plateau through 48 h of treatment with either MG adduct (Appendix A). We postulate that MG adducts may preferentially activate different pathways with different degrees of intensity and duration, which can contribute to its unique cellular effects, depending on which cascade it activates.

## 4. Discussion

The ubiquitous nature of the endothelium in the body contributes to its susceptibility to damage. Endothelial dysfunction is a major precursor underlying many vascular disorders associated with diabetic complications [52,53,54]. In diabetes, many of these complications arise from dysglycemia, causing elevated inflammation and impaired endothelial homeostasis, which together can lead to endothelial dysfunction. Past studies have linked dysregulated metabolism with endothelial dysfunction, such as elevated levels of glucose and MG causing the modification of intracellular and extracellular proteins [55,56,57], leading to the activation of the unfolded protein response, inflammation, and thrombosis in human aortic endothelial cells [58]. However, there remains a gap in knowledge in the underlying biochemical mechanisms driving vascular diseases such as diabetic nephropathy, retinopathy, cardiopathy, and neuropathy. This is a contributing factor to late-stage diagnosis and an increased likelihood for developing secondary complications, increasing the risk of mortality [59]. As by-products of dysregulated metabolism, we propose that over time, exposure of the endothelium to MG adducts in the body may lead to overt vascular disease [60]. In this study, we evaluated nucleoside MG adducts as drivers of vascular dysfunction in HUVECs treated with clinically relevant levels of MG adducts to mimic levels found in patients [9,23]. Herein, we are the first to report that nucleoside MG adducts drive endothelial dysfunction through the activation of RAGE, a finding unique from past studies that demonstrated activation solely with large macromolecular ligands, such as proteins and other AGEs [16,61,62,63]. We recently identified MG adducts as biomarkers for DKD [9]. These studies provide the first biochemical characterization of the mechanisms by which MG adducts may function as not only biomarkers but also drivers of vascular dysfunction and disease.

We found that under diabetogenic conditions, namely hyperglycemia and hyperlipidemia, there is an increase in the intracellular and secreted levels of MG adducts. This suggests that MG adducts may accumulate inside cells and be removed and transported out of the cell as a free nucleoside through a currently unknown mechanism. Further studies into a dose-dependent or time-dependent effect are warranted to understand adduct rates of formation and turnover. We speculate that secreted MG adducts may originate from the repair of adducts formed in genomic material, released during macromolecular breakdown or cell death, or as nucleoside cargo in extracellular vesicles, though the precise mechanism of repair and export remains unclear. In this regard, we postulate that secreted MG adducts may act upon HUVECs and other cells within the endothelial milieu via paracrine and/or autocrine signaling. 

MG adduct exposure in HUVECs resulted in dysfunction primarily mediated through RAGE. We found that additional consequences of RAGE activation included the phosphorylation of AKT and MEK and the nuclear translocation of NFκB p65. Our findings indicated a significant increase in AKT activation via phosphorylation within 1 h of adduct treatment, which may indicate the HUVECs recognizing the MG adducts as damaging external stimuli and activating survival pathways. However, our time course study revealed that this effect dissipated as early as 3 h of adduct treatment, with AKT phosphorylation returning to basal levels. In the context of endothelial dysfunction, the PI3K/AKT family more closely associates with pro-survival pathways [64,65,66,67]. Conversely, the 1-h adduct treatment led to a modest increase in MEK phosphorylation (1.2–1.5 fold), but a significant 3-fold increase in MEK phosphorylation beginning from 3 h of treatment, lasting at least 48 h post-exposure, which may represent long-term detrimental effects, leading to endothelial deficits over time. The RAS–MAPK pathway, which includes MEK, is associated with inflammation, endothelial injury, and vasoconstriction [65,68,69]. This suggests that endothelial cells can activate mechanisms to overcome dysfunction and damaging stimuli, but prolonged exposure may eventually overwhelm their homeostatic mechanisms.

Notably, the pre-treatment of HUVECs with a small molecule inhibitor of intracellular RAGE signaling ablated these signaling events and defects in endothelial function, demonstrating that these effects are primarily mediated through RAGE. However, we postulate this was not a full reversal to basal levels, as there were residual indicators of dysfunction remaining; we speculate this may be due to two possible scenarios. First, the RAGE inhibitor’s function may be transient and not stable, and therefore unable to fully suppress RAGE activity long-term. Alternatively, there may be a secondary mechanism through which MG adducts can induce dysfunction, i.e., via another receptor, and this remains to be elucidated. A recent study identified that AGEs mediate ECD by upregulating the expression of G protein-coupled receptors (GPCRs), a phenomenon that was found to occur independent of RAGE [70]. Therefore, it is of interest to determine whether MG adduct-induced ECD is solely dependent on RAGE, or whether MG adducts can cause ECD through other mechanisms.

Furthermore, our RNA sequencing analysis revealed common mechanisms through which CEdG and CEG elicit endothelial dysfunction, identifying a transcriptomic signature, including upregulated cell death, inflammation, ECM remodeling, stress responses, and angiogenesis, and downregulated cell cycle, endothelial function, cell junction, and development. Both signatures were diminished with the pre-treatment with Ri, and we also identified changes in gene expression that were unique to either CEdG or CEG and not a shared effect. For example, CEdG significantly upregulated the genes involved in ECM remodeling and regulators of EC function and integrity, while CEG upregulated more genes involved in cell death and endothelial invasiveness. Conversely, CEdG significantly downregulated the genes involved in genomic stability and protection against oxidative stress, whereas CEG downregulated the genes involved in the cell cycle and maintenance of EC function. This demonstrates that although both CEdG and CEG appear to primarily act through RAGE, they may lead to the activation of different signaling cascades and endothelial dysfunction via different mechanisms. This may be due to different interactions, RAGE dimer formations, or variable intracellular changes in response to each ligand. The elucidation of unique RAGE-independent mechanisms of MG adduct-induced endothelial dysfunction also warrants further study.

Potential future studies of interest include an in vivo model to validate our in vitro findings, which can be further extended using organ-specific cells such as glomerular endothelial cells. We are particularly interested in studying the impact of MG adduct treatment on vascular disease progression and severity, particularly DKD, and if this is RAGE dependent. RAGE is ubiquitously expressed in different cell types throughout the kidneys, such as the renal endothelium, podocytes, and tubular ducts, and is particularly upregulated during diabetic nephropathy [13,71]. Unfortunately, many canonical biomarkers and measures of vascular disease often do not become clinically relevant until significant vascular damage has already occurred. It is conceivable that early repeated exposure of MG adducts to RAGE in the kidneys can lead to prolonged endothelial dysfunction, causing impairments in the glomerular filtration rate, and enhanced albumin excretion as a result [72,73,74,75,76,77,78,79,80]. Furthermore, it is important to define the binding kinetics and interactions of free nucleoside MG adducts to RAGE, which have not been previously explored. There are potential sites for interaction through the carboxyethyl moiety of MG adducts, including the V-domain of RAGE, which has been previously identified as a ligand binding site for protein AGEs [18,62,81]. These studies will help in our understanding of the spatiotemporal properties of MG adduct-induced RAGE activation in mediating endothelial cell dysfunction. They may also inform on treatment strategies to target and prevent interactions between MG adducts and RAGE, thus preventing vascular dysfunction.

## 5. Conclusions

In summary, the nucleoside MG adducts CEdG and CEG are produced and secreted by endothelial cells into the extracellular matrix under diabetogenic conditions. Once in the extracellular space, we propose that they can act through autocrine or paracrine signaling to activate RAGE. The consequence of this activation is endothelial dysfunction, a phenomenon that is mitigated with pre-treatment with a RAGE inhibitor compound. Sustained exposure to MG adducts and prolonged ECD may be the key to the development of vascular diseases. These studies provide the first biochemical characterization of the mechanisms through which MG adducts may function as not only biomarkers of DKD but also drivers of vascular disease, and identify a novel class of RAGE ligands and potential therapeutic modality to improve patient outcomes.

## Figures and Tables

**Figure 1 antioxidants-13-00085-f001:**
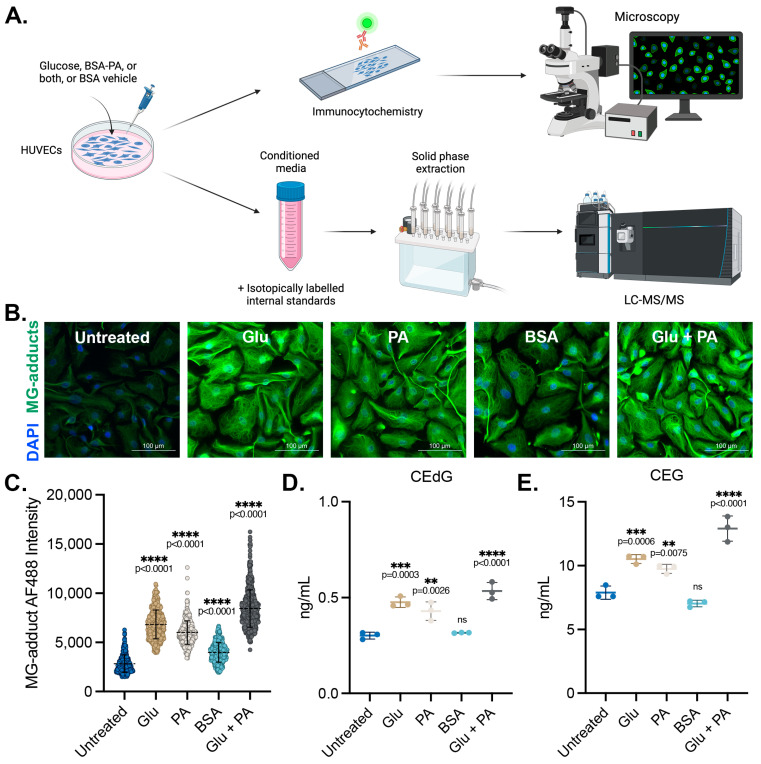
MG adducts are produced intracellularly and secreted by endothelial cells under diabetogenic conditions. (**A**) Design schematic and pipeline. Created with BioRender.com. (**B**) Immunofluorescent analysis of intracellular MG adducts in HUVECs exposed to 25 mM Glu, 100 µM PA, BSA, or 25 mM Glu + 100 µM PA. (**C**) Quantification of (**B**). N = 3; the data represent three independent biological replicates. A minimum of 300 cells per condition were analyzed. Statistical significance was determined via the one-way ANOVA with Dunnett’s multiple comparisons test. Extracellular, secreted levels of (**D**) CEdG and (**E**) CEG were quantified using LC-MS/MS with quantitation to isotopically labeled internal standards. N = 3; data represent three independent biological replicates. Statistical significance was determined via the one-way ANOVA with Dunnett’s multiple comparisons test.

**Figure 2 antioxidants-13-00085-f002:**
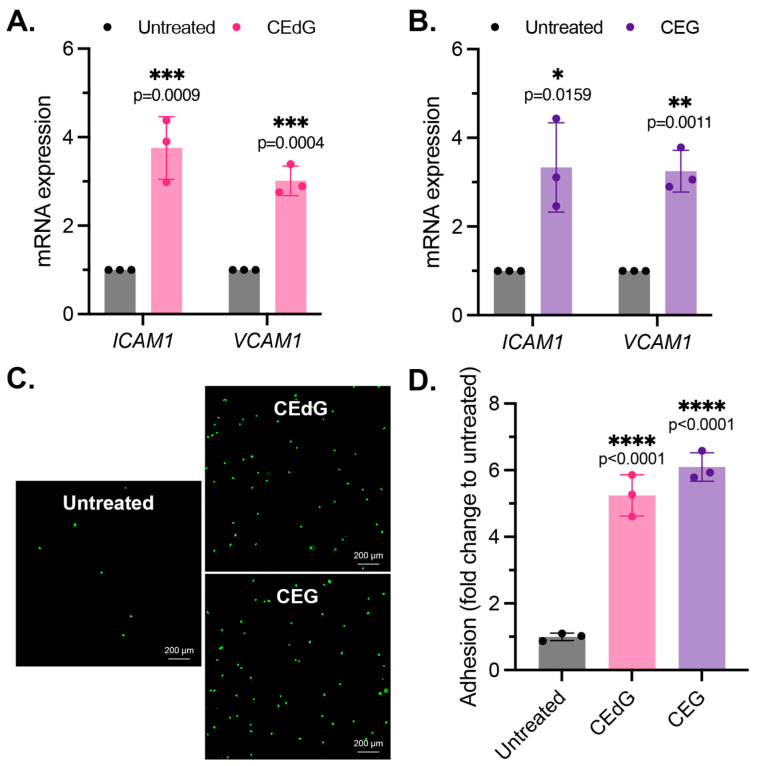
MG adducts activate endothelial cells and promote monocyte adhesion. (**A**,**B**) qPCR analysis of *ICAM1* and *VCAM1* in HUVECs treated with 100 ng/mL of *R*,*S-*CEdG or CEG for 1 h. Statistical significance was determined via the unpaired *t*-test. N = 3; the data represent three independent biological replicates. (**C**) Functional effects on endothelial activation were assessed via the monocyte adhesion assay. HUVECs were treated with 100 ng/mL of *R*,*S-*CEdG or CEG for 1 h. (**D**) Quantification of (**C**). N = 3; the data represent three independent biological replicates. Statistical significance was determined via the one-way ANOVA with Dunnett’s multiple comparisons test.

**Figure 3 antioxidants-13-00085-f003:**
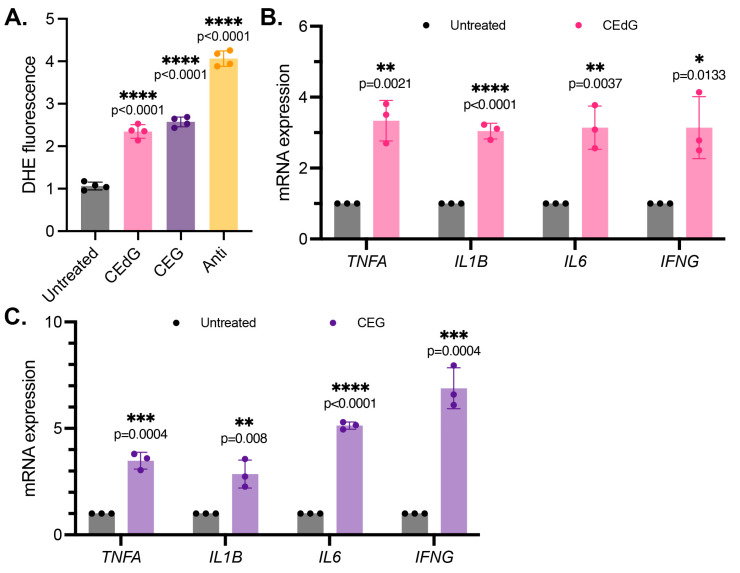
MG adducts upregulate ROS production and expression of pro-inflammatory cytokines in HUVECs. (**A**) ROS were detected via a dihydroxyethidium (DHE) probe in HUVECs treated with MG adducts. HUVECs were treated with 100 ng/mL of *R*,*S-*CEdG or CEG for 1 h. N = 4; the data represent four independent biological replicates with technical triplicates. Statistical significance was determined via the one-way ANOVA with Dunnett’s multiple comparisons test. (**B**,**C**) qPCR analysis of *TNFA*, *IL1B*, *IL6*, and *IFNG* in HUVECs treated with 100 ng/mL of *R*,*S-*CEdG or CEG for 1 h. N = 3; the data represent three independent biological replicates. Statistical significance was determined via the unpaired *t*-test.

**Figure 4 antioxidants-13-00085-f004:**
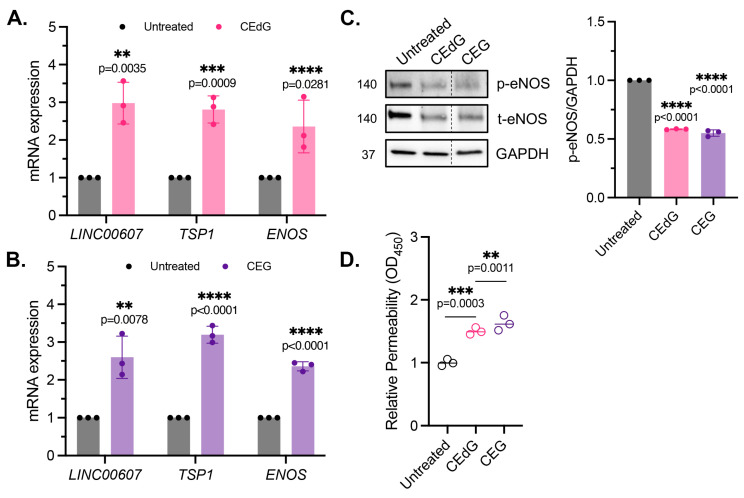
MG adducts induce endothelial dysfunction and impair endothelial homeostasis and barrier integrity. (**A**,**B**) qPCR analysis of *LINC00607*, *TSP1*, and *ENOS* in HUVECs treated with 100 ng/mL of *R,S-*CEdG or CEG for 1 h. N = 3. Statistical significance was determined via the unpaired *t*-test. (**C**) Western blot analysis for phosphorylated and total eNOS in HUVECs treated with 100 ng/mL of *R*,*S-*CEdG or CEG for 1 h. N = 3. Statistical significance was determined via the one-way ANOVA with Dunnett’s multiple comparisons test. (**D**) Endothelial permeability of HUVECs treated with 100 ng/mL of *R*,*S-*CEdG or CEG for 6 h. N = 3. Statistical significance was determined via the one-way ANOVA with Dunnett’s multiple comparison test.

**Figure 5 antioxidants-13-00085-f005:**
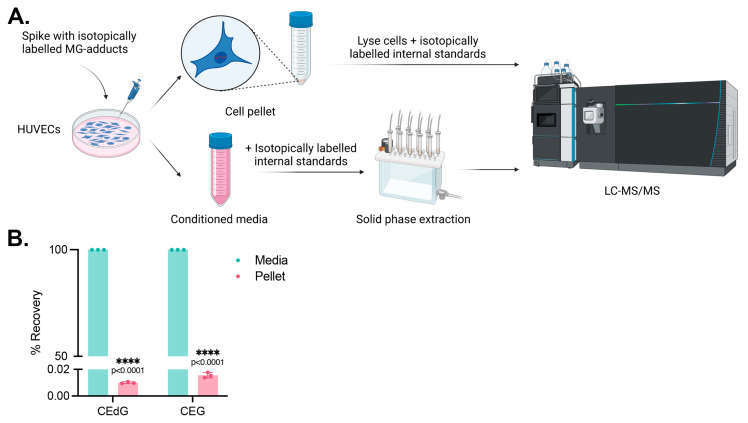
MG adducts are not taken up by HUVECs. (**A**) Design schematic and pipeline. Created with BioRender.com. (**B**) HUVECs were treated with 200 ng/mL isotopically labeled *R*,S-CEdG or CEG for 24 h. The conditioned media and cell pellet were harvested and processed, and the levels of MG adducts in each compartment were quantified via LC-MS/MS with quantitation to isotopically labeled internal standards. N = 3; the data represent three independent biological replicates. Statistical significance was determined via the unpaired *t*-test.

**Figure 6 antioxidants-13-00085-f006:**
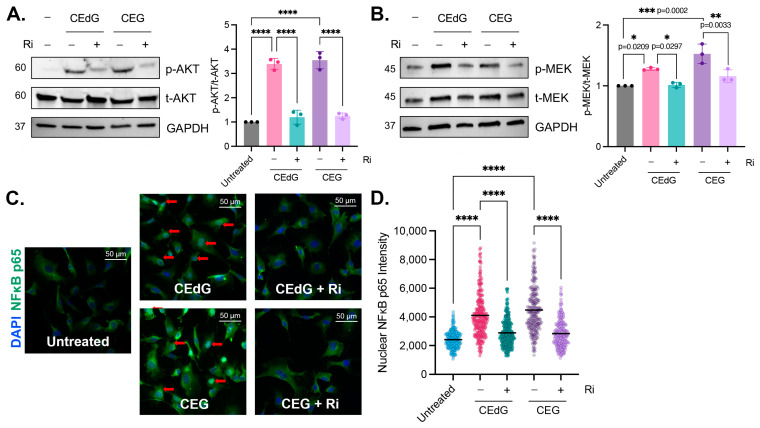
MG adducts activate RAGE signaling and promote NFκB p65 nuclear translocation. Western blot analysis of total and phosphorylated (**A**) AKT and (**B**) MEK. HUVECs were treated with 100 ng/mL of *R*,*S-*CEdG or CEG for 1 h, with or without 100 µM Ri pretreatment for 1 h. N = 3; the data represent three independent biological replicates. Statistical significance was determined via the one-way ANOVA with Tukey’s multiple comparisons test. **** *p* < 0.0001. (**C**) NFκB p65 nuclear translocation was measured via immunofluorescence and brightfield microscopy. HUVECs were treated with 100 ng/mL of *R*,*S-*CEdG or CEG for 1 h, with or without 100 µM Ri pretreatment for 1 h. (**D**) Quantification of (**C**). N = 3; the data represent three independent biological replicates. Statistical significance was determined via the one-way ANOVA with Tukey’s multiple comparisons test. **** *p* < 0.0001.

**Figure 7 antioxidants-13-00085-f007:**
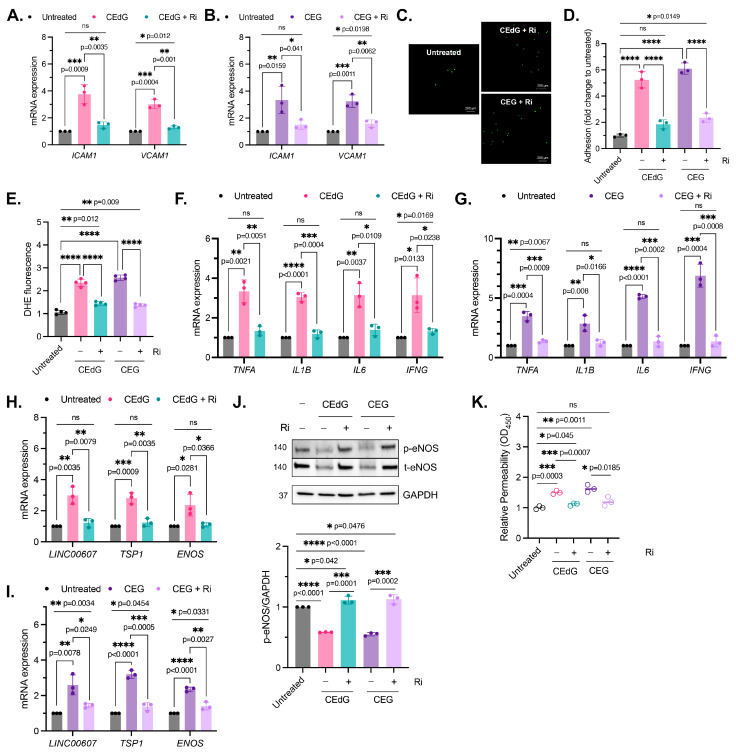
MG adduct-induced endothelial dysfunction is mediated through RAGE signaling. (**A**,**B**) qPCR analysis of *ICAM1* and *VCAM1* in HUVECs pre-treated with 100 µM Ri for 1 h, followed by 100 ng/mL of *R*,*S-*CEdG or CEG for 1 h. Statistical significance was determined via multiple unpaired *t*-tests. N = 3; the data represent three independent biological replicates. Untreated, CEdG-treated, and CEG-treated samples were run in tandem and are derived from Figure 2A,B, and are included here for the purpose of the comparison with RAGE inhibitor-treated samples. (**C**) Functional effects on endothelial activation were assessed via the monocyte adhesion assay. HUVECs were pre-treated with 100 µM Ri for 1 h, followed by 100 ng/mL of *R*,*S-*CEdG or CEG for 1 h. (**D**) Quantification of (**C**). N = 3; the data represent three independent biological replicates. Statistical significance was determined via the one-way ANOVA with Tukey’s multiple comparisons test. Untreated, CEdG-treated, and CEG-treated samples were run in tandem and are derived from Figure 2D, and are included here for the purpose of the comparison with RAGE inhibitor-treated samples. (**E**) ROS were detected via the dihydroxyethidium (DHE) probe in HUVECs pre-treated with 100 µM Ri for 1 h, followed by 100 ng/mL of *R*,*S-*CEdG or CEG for 1 h. N = 4; the data represent four independent biological replicates with technical triplicates. Statistical significance was determined via the one-way ANOVA with Tukey’s multiple comparisons test. Untreated, CEdG-treated, and CEG-treated samples were run in tandem and are derived from Figure 3A, and are included here for the purpose of the comparison with RAGE inhibitor-treated samples. qPCR analysis of (**F**,**G**) *TNFA*, *IL1B*, *IL6,* and *IFNG*, (**H**,**I**) *ENOS*, *LINC00607*, and *TSP1* in HUVECs pre-treated with 100 µM Ri for 1 h, followed by 100 ng/mL of *R*,*S-*CEdG or CEG for 1 h. N = 3; the data represent three independent biological replicates. Statistical significance was determined via multiple unpaired *t*-tests. Untreated, CEdG-treated, and CEG-treated samples were run in tandem and are derived from Figure 3B,C and Figure 4A,B, and are included here for the purpose of the comparison with RAGE inhibitor-treated samples. (**J**) Western blot analysis for phosphorylated and total eNOS in HUVECs pre-treated with 100 µM Ri for 1 h, followed by 100 ng/mL of *R*,*S-*CEdG or CEG for 1 h. N = 3. Statistical significance was determined via the one-way ANOVA with Tukey’s multiple comparisons test. Untreated, CEdG-treated, and CEG-treated samples were run in tandem and are derived from Figure 4C, and are included here for the purpose of the comparison with RAGE inhibitor-treated samples. (**K**) Endothelial permeability of HUVECs treated with 100 ng/mL of *R*,*S-*CEdG, or CEG for 6 h, with or without pre-treatment of 100 µM Ri for 1 h. N = 3. Statistical significance was determined via the one-way ANOVA with Tukey’s multiple comparison test. Untreated, CEdG-treated, and CEG-treated samples were run in tandem and are derived from Figure 4D, and are included here for the purpose of the comparison with RAGE inhibitor-treated samples. **** *p* < 0.0001.

**Figure 8 antioxidants-13-00085-f008:**
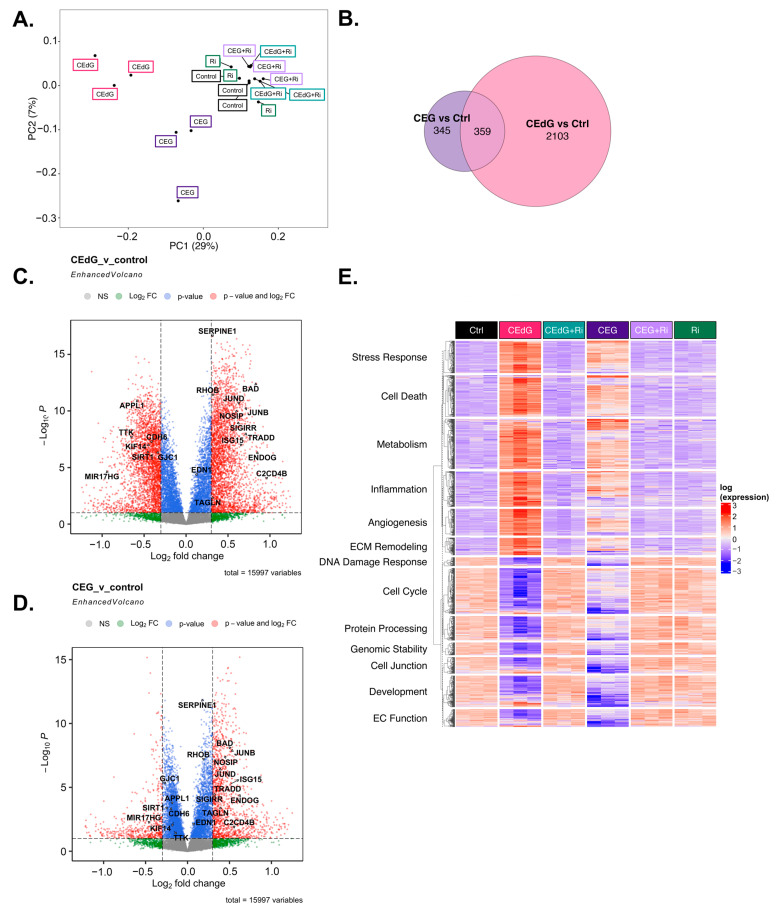
Transcriptomic effect of MG adducts on endothelial cells. (**A**) Principal component analysis of treated HUVEC samples, where replicates are colored according to their respective sample types. (**B**) Venn diagram highlighting the DEGs that meet a threshold of p.adj < 0.1 and log_2_FC of ±0.5 in treated HUVECs. Volcano plot representing a selection of genes and highlighting their fold change and *p*-value in cells treated with (**C**) CEdG or (**D**) CEG. (**E**) Gene heatmap constructed from GO-BP pathways associated with endothelial function. DEGs with p.adj < 0.1 are shown.

**Figure 9 antioxidants-13-00085-f009:**
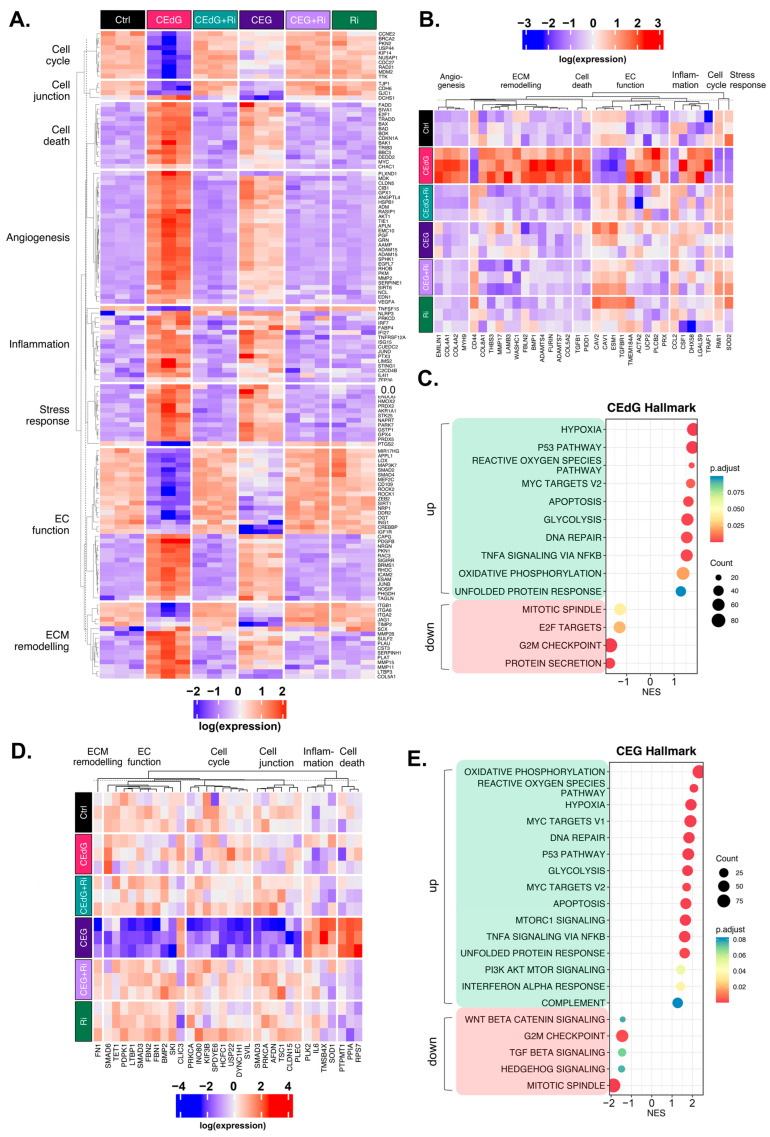
Unique and common changes in gene and pathway expression in HUVECs treated with MG adducts. (**A**) Annotated heatmap highlighting the genes involved in the cell cycle, cell death, cell junction, angiogenesis, inflammation, stress responses, EC function, and ECM remodeling, and their change in expression in response to exposure to either CEdG or CEG, with or without Ri pre-treatment. Common DEGs that are significant (p.adj < 0.1) in both CEdG and CEG are shown. (**B**) DEGs uniquely differently regulated by treatment with CEdG. (**C**) Hallmark pathway analysis for HUVECs treated with 100 ng/mL of *R*,*S-*CEdG for 1 h. (**D**) DEGs uniquely differently regulated by treatment with CEG. (**E**) Hallmark pathway analysis for HUVECs treated with 100 ng/mL of *R*,*S-*CEG for 1 h.

**Figure 10 antioxidants-13-00085-f010:**
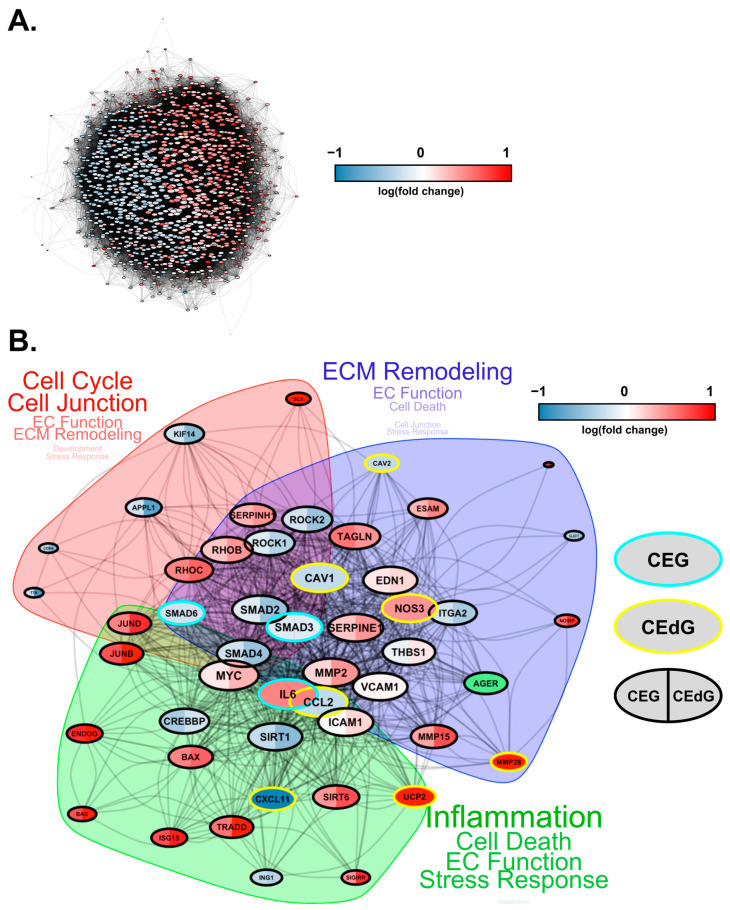
Protein–protein interaction network for DEGs impacted by CEdG and CEG. (**A**) Protein interaction network (STRING DB) showing the interactions among the differentially expressed genes in HUVECs treated with CEdG or CEG. The nodes represent proteins, and the edges connecting the nodes represent interactions. The node size is proportional to the number of interactions, while the color is according to the average log fold-change in CEdG vs the control and CEG vs the control. The RAGE gene is highlighted in green. (**B**) A smaller subnetwork showing the interactions among select DE genes. Nodes (DE genes) that are common between CEdG and CEG are highlighted with black borders. The common nodes are colored using split coloring, where the left half is colored according to the log fold-change in CEG vs the control, while the right half according to CEdG vs the control. Nodes with cyan and yellow borders represent DEGs unique to CEG and CEdG, respectively. The protein interaction network was clustered into three regions, as highlighted by the shaded areas. The key cellular processes represented by each cluster are highlighted next to each cluster, with the size of the text proportional to the number of genes involved in each process.

## Data Availability

The transcriptomic data presented in this study are openly available in the National Center for Biotechnology Information (NCBI)’s Gene Expression Omnibus (GEO), under accession number GSE251645.

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
