# Peer review of "Methylglyoxal-Derived Nucleoside Adducts Drive Vascular Dysfunction in a RAGE-Dependent Manner"

_antioxidants, 2024, doi:10.3390/antiox13010085_

Round 1
Reviewer 1 Report
Comments and Suggestions for Authors
The authors showed that endothelial dysfunction along with enhanced monocyte adhesion, increased reactive oxygen species production, endothelial permeability is caused by the most common nucleoside adducts, which are N2-(1-carboxyethyl) -deoxyguanosine (CEdG) and N2-(1-carboxyethyl) -guanosine (CEG). They found that the endothelial dysfunction were mediated by the RAGE and found that inhibition of the intracellular RAGE signaling attenuated these abnormal inflammatory changes.
This study was well conducted and provided crucial insights of the molecular mechanism of endothelial dysfunction.
The figure 10 (protein-protein interaction network for DEGs impacted by CEdG and CEG) indicated that smad 2, 3 and 4 are placed in the central position in this network. Since the phosphorylations of those R-Smads are critical for tissue fibrosis leading to the development of DKD. Therefore, the reviewer would like to know whether CEdG or CEG modulates phosphorylations of the R-Smads in HUVECs.
Reviewer 2 Report
Comments and Suggestions for Authors
In this paper, the authors investigate the molecular mechanisms by which methylglyoxal-derived nucleosides adducts may cause vascular dysfunction in vitro, demonstrating that these effects are mediated by RAGE. On the whole, the experimental design is well conducted, the results are rich and robust, and support the conclusions.
My MAIN CONCERNS are the following:
Results
Paragraph 3.1: Please specify glucose concentration for untreated cells (fig. 1). Also, it would be useful to report whether you also performed glucose dose-response studies, even if you do not show the results.
Most results show gene expression studies, rather than protein expression studies, and both expressions do not necessarily parallel. Please discuss.
Discussion
Lines 674- 676: the authors could cite pertinent papers showing the relationship between dysglycemia and inflammation, adhesion molecules, leading to endothelial dysfunction (i.e. King DE, et al, Diabetes Care 2003; Palella E., et al., Int J Environ Res Public Health 2020; Vissoky Cé, G., et al. J Clin Endocrinol Metab 2011).
The authors have previously published that methylglyoxal derivatives may predict DKD risk in type 1 diabetic patients. However, as this is an in vitro study, I would emphasize a generalized potential role in endothelial dysfunction and soften references to DKD (lines 691-693).
MINOR POINTS:
Line 102: THP-1 should be specified.
Line 393: Thromospondin-1 should be Thrombospondin-1, please fix.
Round 2
Reviewer 1 Report
Comments and Suggestions for Authors
I have no further comments.